# Analyzing HPV Vaccination Service Preferences among Female University Students in China: A Discrete Choice Experiment

**DOI:** 10.3390/vaccines12080905

**Published:** 2024-08-09

**Authors:** Lu Hu, Jiacheng Jiang, Zhu Chen, Sixuan Chen, Xinyu Jin, Yingman Gao, Li Wang, Lidan Wang

**Affiliations:** 1School of Health Management, Anhui Medical University, Hefei 230032, China; hulu89@stu.ahmu.edu.cn (L.H.); 2245010590@stu.ahmu.edu.cn (J.J.); 2345010652@stu.ahmu.edu.cn (Z.C.); 2345010621@stu.ahmu.edu.cn (S.C.); 2345010642@stu.ahmu.edu.cn (X.J.); gaoyingman20020214@163.com (Y.G.); 2Centre for Health Policy Research, Anhui Medical University, Hefei 230032, China

**Keywords:** female university students, HPV vaccination service, health preference, discrete choice experiment, China

## Abstract

Objective: Despite being primary beneficiaries of human papillomavirus (HPV) vaccines, female university students in China exhibit low vaccination rates. This study aimed to assess their preferences for HPV vaccination services and evaluate the relative importance of various factors to inform vaccination strategy development. Methods: Through a literature review and expert consultations, we identified five key attributes for study: effectiveness, protection duration, waiting time, distance, and out-of-pocket (OOP) payment. A D-efficient design was used to create a discrete choice experiment (DCE) questionnaire. We collected data via face-to-face interviews and online surveys from female students across seven universities in China, employing mixed logit and latent class logit models to analyze the data. The predicted uptake and compensating variation (CV) were used to compare different vaccination service scenarios. Results: From 1178 valid questionnaires, with an effective response rate of 92.9%, we found that effectiveness was the most significant factor influencing vaccination preference, followed by protection duration, OOP payment and waiting time, with less concern for distance. The preferred services included a 90% effective vaccine, lifetime protection, a waiting time of less than three months, a travel time of more than 60 min, and low OOP payment. Significant variability in preferences across different vaccination service scenarios was observed, affecting potential market shares. The CV analysis showed female students were willing to spend approximately CNY 5612.79 to include a hypothetical ‘Service 5’ (a vaccine with higher valency than the nine-valent HPV vaccine) in their prevention options. Conclusions: The findings underscore the need for personalized, need-based HPV vaccination services that cater specifically to the preferences of female university students to increase vaccination uptake and protect their health.

## 1. Introduction

Cervical cancer, primarily caused by the human papillomavirus (HPV) infection, remains a significant global health concern. Persistent infection with high-risk HPV types, especially type 16, is a known cause of cervical cancer [1]. According to World Health Organization (WHO) data from 2020, there were approximately 604,000 new cases and 343,000 deaths due to cervical cancer worldwide [2]. Notably, Chinese women accounted for about 110,000 of these new cases—representing approximately 18.3% of the global total—and around 60,000 deaths, or 17.3% of the worldwide fatalities [3]. HPV vaccination can effectively prevent HPV infection and reduce the incidence of related diseases [4]. Currently, bivalent (2vHPV), quadrivalent (4vHPV), and nine-valent HPV (9vHPV) vaccines have been approved in the global market for the prevention of cervical cancer caused by HPV infection [5,6]. Foreign 2vHPV, 4vHPV and 9vHPV vaccines were approved in 2006, 2007, and 2014, respectively [7]. China’s 2vHPV vaccine was approved for the market in 2019 and 2022 [8]. These vaccines are highly effective in preventing infections and related diseases caused by the HPV types they cover. The 2vHPV vaccine targets HPV types 16 and 18, which are responsible for 70% of all cervical cancer cases. The 4vHPV vaccine additionally covers HPV types 6 and 11, accounting for 90% of all genital warts cases. The most comprehensive option, the 9vHPV vaccine, includes protection against HPV types 6, 11, 16, 18, 31, 33, 45, 52, and 58, collectively responsible for 90% of genital warts and cervical cancer cases [9,10].

The WHO Global Strategy aims to eliminate cervical cancer as a public health issue, with ambitious targets set for 2030 to guide global efforts. These include vaccinating 90% of girls with the HPV vaccine by age 15, screening 70% of women with a high-quality test by ages 35 and 45, and ensuring that 90% of women with diagnosed cervical disease receive treatment. This strategy seeks to reduce the incidence of new cervical cancer cases to fewer than 4 per 100,000 women annually [2]. By the end of 2020, HPV vaccines were used in 129 countries to prevent HPV-related diseases, and 111 of these countries have included HPV vaccines in their national immunization programs [11,12]. Indeed, the proportions of HPV vaccination among the appropriate age group (9–17 years old) in the national immunization program were found to be high [13]. For example, 75.1% of adolescents aged 13–17 in the United States received at least one dose of HPV vaccine [14]. In Australia, 90.9% of 15-year-old girls received one dose of HPV vaccine [15]. However, the HPV vaccine has not been included in the national immunization program of China [16], with less than 6% of the appropriately aged population receiving the vaccine [17]. Moreover, previous studies in China found that female university students had low HPV vaccination tendencies [18,19,20,21].

Existing researches on the low vaccination rates of female students discovered that a high level of cognition regarding HPV and HPV vaccines, history of sexual behavior, history of cancer in family or friends, age, grade, education level of parents, and personal monthly living expenses have an impact on improving the HPV vaccination rate. They all focused on the demographic characteristics which influenced students’ choices of HPV vaccination services. The discrete choice experiment (DCE) is an econometric method for quantifying preferences and has been applied in many fields [22]. The DCE results can assist policymakers in understanding which characteristics or features of public health programs have the highest preferences [23,24]. Existing studies have widely applied DCE to investigate preferences for different vaccines, such as the COVID-19 vaccination [25] and infant meningococcal vaccination [26]. However, few studies have applied this method to identify the factors influencing female university students’ preferences for HPV vaccination. They have instead mainly focused on vaccine characteristics such as effectiveness, protection duration, side effects, and vaccination age [27,28], while some have chiefly considered vaccine service attributes such as waiting time for appointments and service time [29]. Moreover, these were small-sample studies centered on a single city, and only incorporated parts of our country, such as Shandong, Hong Kong, and Zhejiang.

In this study, focus was on both vaccine characteristics and vaccine service attributes, and the scope of our study was extended to all of China, meaning that the general applicability of our findings to the whole country may be stronger than that of the above-mentioned works. Based on those aforementioned studies, we explored the following problems in depth: (a) What factors do female students consider when balancing their vaccination options? (b) What is the relative importance (RI) of these factors? (c) What are the different preferences of different subgroups of female students? The answers to these questions are important when it comes to adequately understanding female students’ preferences and exploring the key factors of the low vaccination rate of female students, which will be helpful in improving the vaccination rate and protecting the health of female students.

## 2. Methods

### 2.1. Identification of Attributes and Levels

DCE design and analysis were based on the checklist and report of the International Society for Pharmacoeconomics and Outcomes Research (ISPOR) Conjoint Analysis Task Forces [22,30,31]. An initial set of attributes was derived from a literature review on the application of DCE in HPV vaccine-related research. The literature review of previous research revealed a few attributes identified as important factors. We initially identified 11 attributes, following which we invited experts in the field of health policy to conduct semi-structured interviews and assessed the appropriateness of the attributes and their levels. Ultimately, based on the literature review and interview results, we settled on effectiveness, protection duration, waiting time, distance, and out-of-pocket (OOP) payment, along with their respective levels (Table 1).

### 2.2. Survey Design

Based on the given attributes and level settings, 768 (4^4^ × 3^1^) pairs of choice sets may exist. However, the use of a full factorial design is inconsistent with the actual situation [32]. Therefore, a partial factorial design was employed to create an efficient design using Stata 16.0 to maximize the D-efficiency [30], and 14 DCE choice sets were finally determined.

Each choice set contained two hypothetical HPV vaccination services and an opt-out option (Table 2). An opt-out option was included to reduce the risk of overestimating attribute influence [33,34]. Students were asked to choose their preferred option among service A, service B and opt-out.

We set up a warm-up set at the first stage so that the students could become familiar with it as soon as possible, while we also installed a test choice set at the tenth stage to test the correctness and validity of students’ answers to the questionnaire (Table 2). In both tasks, all attribute levels of service A were superior to those of service B. The students were considered to have failed the test if they did not choose service A [35]. To ensure that enough valid online questionnaire data were collected [36], alongside keeping the content of the 16 choice sets consistent with the face-to-face survey questionnaire, we added an additional arithmetic question to the online survey questionnaire.

Besides the DCE section, the questionnaire mainly included demographic characteristics of the students, such as major, grade, age, type of household registration, and education level of parents. The type of household registration in China is known as Hukou status, which is a public certificate book that registers and certifies a natural person according to the household and records the natural person’s name, date of birth, relatives, marital status, etc. Currently, there are two main types of registration: urban and rural household registration [37,38,39].

### 2.3. Sampling

A multistage random sampling method was used to select a representative sample of female students in China (Figure 1). First, we randomly chose areas in East, North, Central, South, Northwest, Southwest, and Northeast China. We then selected universities from these areas. Finally, seven universities were identified, including four medical universities and three comprehensive universities (Figure 2).

Affected by COVID-19, this study was conducted between October and November 2022 using a combination of face-to-face and online questionnaires. Respondents who met the following criteria were included in this study: (a) they are female, (b) they are students at university, and (c) they have the ability to give informed consent and autonomy. The sample size was determined according to Ohm’s rule:
$N>500\times \frac{c}{t\times a}$ [40], where the largest number of levels, *c*, among the different attributes in this study was 4, while the number of choice sets, *t*, and the number of alternatives in each set, *a*, were 14 and 3, respectively. To ensure representativeness of the sample size and comparability of grades and majors, the sample size of each university was determined to be at least 150 students.

We selected one to three university students from each target university as investigators for the questionnaire survey. Then, we conducted strict training for them in accordance with a standard survey manual. The investigators selected students from at least 2 majors and 2 grades in each university, and then summoned the students to the classroom with the assistance of the counselors for the questionnaire survey.

### 2.4. Piloting and Formal Investigation

The whole investigation was divided into two stages: pre-investigation and formal investigation. A sample of 30 female students from each university was investigated to help surveyors become familiar with the questionnaire content and check the comprehensibility, acceptability, and effectiveness of the questionnaire, and the existing problems in the pre-investigation were further modified.

After successful completion of the pre-investigation, a formal investigation was conducted at seven universities in six provinces of China. During the investigation, each investigator provided one-on-one guidance to each participant in filling out the questionnaire. Each investigator worked on tasks using survey manual. The manual mainly included the following: (a) the significance of the investigation, (b) the structure of the questionnaire and the detailed definition of each attribute of the DCE, and (c) the instructions on how to help participants understand the choice sets through the warm-up set. The manual also provided an explanation of the investigation steps, including the instructions for terminating the session if the participant showed signs of not understanding the choice tasks or had trouble completing them.

In the formal investigation, 963 pieces of online data and 305 face-to-face surveys data were collected. A set of strict quality control standards was formulated to filter the data. If a piece of data met any of the following criteria, then it was excluded and not included in the final data analysis: (1) filled in the same answer for all 16 choice sets, (2) the arithmetic problem was miscalculated, and (3) the warm-up set and test choice set were filled out incorrectly. Finally, we retained 1178 valid data, including 899 online survey data and 279 offline survey data.

### 2.5. Statistical Analysis

#### 2.5.1. Model Specification

A mixed logit model was employed to estimate students’ preferences for HPV vaccination services. The parameters of the attribute levels are assumed to conform to a normal distribution. In the model, OOP payment was included as a continuous variable, while other attribute levels were introduced as categorical variables using dummy coding [41]. Additionally, the model included an alternative-specific constant (ASC), representing the utility generated by the opt-out option.

To ensure reliable parameter estimation, we iteratively estimated the mixed logit model, starting with 50 iterations and gradually increasing the random draws in increments of 500. Model fit was assessed using the Akaike Information Criterion (AIC) and the Bayesian Information Criterion (BIC) as benchmarks [42]. Following multiple iterations, the model exhibited relatively optimal fit results when the number of iterations reached 4000 (Appendix A). Hence, we selected these iteration results as the final estimated model.

We excluded students with missing values and conducted a sensitivity check (Appendix A ) to determine whether excluding these students would have a significant impact on the results of the mixed logit model.

#### 2.5.2. Attribute Relative Importance

To determine whether or not each attribute represented the total utility of the program design, we calculated the proportion of each attribute for which the RI was the sum of its utility ranges, so as to gauge the difference which each attribute could represent in the total utility of the program design [43]. The formula used is as follows:
$$RI_{k}=\left(\frac{A_{k}}{\sum_{k=1}^{5}A_{k}}\right)\times 100\%$$

We then estimated the mean RI for each class and the population mean to observe the RI of attributes among the students in each class and the RI of attributes among the overall population.

#### 2.5.3. Subgroup Analysis

In order to investigate differences in students’ preferences, a subgroup analysis was conducted to compare the RI of vaccination service attributes among students with different characteristics. A latent class logit (LCL) model was employed to assess the preference heterogeneity among students [44]. The number of classes was determined according to BIC [45].

#### 2.5.4. Scenario Analysis

We employed two approaches to compare different HPV vaccination services. Firstly, we calculated the predicted uptake, which represents the probability of each service being chosen by students when presented with a given choice set, using the following formula:
$$\mathrm{Pr}\left(choice=j^{*}\right)=\frac{e^{V_{j^{*}}}}{\sum_{j=1}^{J}e^{V_{j}}}$$ where the probability of choosing option, *j**, is determined by the utility level derived from each option, *V_j_*, in the choice set consisting of the *J* option.

The uptake probabilities were derived from the simulated distributions of random coefficients. We rescaled the median values from each simulated uptake distribution by a common factor to ensure that the sum of probabilities within a choice set equaled one. We analyzed changes in market share by varying the relative prices among different services to explore their competitive relationships.

The second approach involves calculating compensating variation (CV). CV is used to measure how much money needs to be provided or taken away from students when HPV vaccination service change, in order to keep them at their initial level of utility. The results of CV can demonstrate how much students are willing to pay additionally for the improvement in vaccination service while maintaining their utility levels constant. The formula is as follows:
$$CV=\frac{1}{\lambda }\left[\ln\overset{J_{0}}{\underset{j=1}{\sum }}e^{V_{j}^{0}}-\ln\overset{J_{1}}{\underset{j=1}{\sum }}e^{V_{j}^{1}}\right]$$ where *λ* represents the marginal utility of OOP payment;
$V_{j}^{0}$ and
$V_{j}^{1}$ are the value of the representative indirect utility function for HPV vaccination services in each choice set before and after changes in service-related quality, respectively; *j* represents the number of choice options in the choice task.

## 3. Results

### 3.1. Demographic Information

In total, 1178 female students were included in the study, yielding an effective rate of 92.9%. The characteristics of the participants are shown in Table 3, with the majority being undergraduates (95.7%) and from comprehensive universities (53.8%), exhibiting an average age of (22.03 ± 0.07) years old. Moreover, 52.0% of the female students mainly came from urban areas and 35.1% had less than CNY 50,000 of annual per capita household income. Moreover, 44.2% of the female students had CNY 1000–1500 of monthly living expenses, and their parents’ educational levels were below junior high school (35.0% and 44.2%, respectively). In addition, 78.3% of participants had no sexual history.

**Table 3 vaccines-12-00905-t003:** Demographic characteristics of respondents (N = 1178).

Variables	Group	n	%
University type	Medical university	544	46.2
	Comprehensive university	634	53.8
Grade	Bachelor’s degree	1127	95.7
	Graduate’s degree	51	4.3
Type of household registration ^a^	Urban area	613	52.0
	Rural area	565	48.0
Annual per capita household income (CNY in thousand)	<50	414	35.1
	50~99	322	27.3
	100~149	203	17.2
	150~199	134	11.4
	≥200	105	8.9
Monthly living expenses (CNY)	<1000	136	11.5
	1000~1500	521	44.2
	1500~2000	349	29.6
	≥2000	172	14.6
The educational level of father	≤Junior high school’s degree	412	35.0
	High school’s degree	363	30.8
	College’s degree	369	31.3
	≥Graduate’s degree	34	2.9
The educational level of mother	≤Junior high school’s degree	521	44.2
	High school’s degree	333	28.3
	College’s degree	299	25.4
	≥Graduate’s degree	25	2.1
Sex history	Yes	256	21.7
	No	922	78.3

Note: ^a^ it is known as Hukou status in China, a public certificate book that registers and certifies a natural person according to their household, and records the natural person’s name, date of birth, relatives, marital status, etc. Currently, there are two main types of urban and rural household registration. CNY: Chinese Yuan.

### 3.2. Preference

The results of the mixed logit model (Table 4) showed that all five attributes included in the study had a significant influence on the students’ preferences for HPV vaccination services. As shown in Table 4, the coefficient of effectiveness, on different levels, showed that the positive influence of effectiveness with 90% against both cervical cancer and genital warts (β = 3.518 > 1.712, *p* < 0.001) was greater than that of effectiveness with 90% against genital warts and 70% against cervical cancer (β = 1.712, *p* < 0.001), compared with effectiveness with 70% only against cervical cancer. The coefficient of protection duration with different levels showed that the positive influence of protection duration on lifetime (β = 2.422 > 1.756, 1.074, *p* < 0.001) was better than 40 (β = 1.756, *p* < 0.001) and 20 years (β = 1.074, *p* < 0.001), as opposed to 10 years. Similarly, the negative coefficient of waiting time with different levels showed that the female students highly valued a waiting time of less than 3 months (β = −0.398 < −0.107, *p* < 0.001). Female students also had strong preferences for a service that requires more than a 60 min drive by public transport to reach the vaccination site (β = 0.329 > 0.135, *p* < 0.001) and low OOP payment. When the vaccination service was more effective, the protection duration was longer, waiting time was shorter, distance was longer, OOP payment was lower, and students’ option intentions were higher.

**Table 4 vaccines-12-00905-t004:** Preferences of female university students in mixed logit model (N = 1178).

Attributes	Levels	Est	SE
Asc		9.818 ***	0.995
Effectiveness	Normal ^c^ (ref)		
	Good ^d^	1.712 ***	0.049
	Very good ^e^	3.518 ***	0.091
Protection duration (year)	10 (ref)		
	20	1.074 ***	0.045
	40	1.756 ***	0.062
	Lifetime	2.422 ***	0.079
Waiting time ^a^ (month)	<3 (ref)		
	3~6	−0.269 ***	0.038
	6~9	−0.107 **	0.044
	9~12	−0.398 ***	0.044
Distance ^b^ (minute)	<15 (ref)		
	15~30	−0.037	0.040
	30~60	0.135 **	0.045
	≥60	0.329 ***	0.040
OOP payment (CNY)		−1.49 × 10^−4^ ***	0.000

Note: ^a^ refers to the waiting time between appointment and successful vaccination of various types of cervical cancer vaccine; ^b^ refers to the driving time it takes to reach the vaccination site by public transport; ^c^ prevents cervical cancer only and the prevention effectiveness is about 70%; ^d^ prevents cervical cancer and genital warts, where the prevention effectiveness of cervical cancer is about 70% and that of genital warts is about 90%; ^e^ indicates that it has a preventive effect on cervical cancer and genital warts, and both prevention effectiveness are about 90%. ref: reference; Est: estimate; SE: standard error; CNY: Chinese Yuan; OOP: out-of-pocket. ** *p* < 0.05; *** *p* < 0.001.

### 3.3. Preference Heterogeneity

Significant preference differences existed between different subgroups. An LCL model was initially used to conduct subgroup analyses on individual demographic factors, revealing that university type, grade, type of household registration, annual per capita household income, monthly living expenses, parental educational level, and sexual life status significantly influenced students’ preferences. According to the BIC, the study was confined to two classes (Table 5). Class 1 accounted for 44.7%, while Class 2 accounted for 55.3%.

Although the two groups exhibited similar results, there were some notable differences. In comparison with the general population, the coefficient of effectiveness (β = 4.860 > 1.498, *p* < 0.001) on different levels showed that the effectiveness seemed to be more important for Class 1 than Class 2. However, Class 1 paid little attention to the protection duration (β = 1.426 < 2.418, *p* < 0.001) compared with Class 2. Students in both Class 1 and Class 2 preferred vaccination services with short waiting times. Students in Class 1 preferred services that were close by, while Class 2 preferred those that were further away. In addition, Class 1 students were less particular about OOP payment of service (β = 5.02 × 10^−4^ > −0.24 × 10^−4^, *p* < 0.001), whereas Class 2 students preferred lower service payment. Class 1 can be considered the budget-rich type, and the other one could be considered the frugal type.

### 3.4. Membership Analysis

The expected values of the significant predictors are shown in Figure 3. In Class 1, 80.0% of female students had no sexual history, while 62.8% of female students’ mothers had a high educational level and 71.2% of female students’ fathers had a high educational level. Furthermore, 90.2% of female students had more than CNY 1000 in monthly living expenses compared with the other class, while 71.1% of female students had an average annual household income higher than CNY 50,000. Class 2 included more comprehensive university students, and 51.4% of students had a rural household registration.

### 3.5. Relative Importance (RI)

We derived the RI for each attribute for the population and two classes, as shown in Figure 4. For the population, effectiveness was the most important attribute (45.4%). The second was protection duration (31.3%), followed by OOP payment (13.5%) and waiting time (5.1%). In contrast, the students did not care much about distance (4.7%).

The students of Class 1 considered effectiveness to be the most important attribute among the five attributes (64.2%), and did not care much about OOP payment (0.05%). In Class 2, the degree of preference for protection duration was the highest (41.7%), and the importance of distance was the lowest (3.4%).

### 3.6. Scenario Analysis

Based on the HPV vaccines currently available in the market, and considering attributes including the distance to the vaccination site, we defined Service 1 as a domestic 2vHPV vaccination service, Service 2 as an imported 2vHPV vaccination service, Service 3 as an imported 4vHPV vaccination service, Service 4 as an imported 9vHPV vaccination service, and Service 5 as a hypothetical vaccination service offering a vaccine with higher valency than the 9vHPV vaccine. Through the results of scenario analysis (Table 6), we found that when comparing domestic and imported 2vHPV services, imported 4VHPV and 9vHPV vaccination services covered the market, and more than half (90.1%) of the students would choose imported 9vHPV vaccination services, though imported 9vHPV vaccination services had the highest OOP payment. On the other hand, the other three vaccine services had less than a 10% market share only.

Notably, when the effectiveness and protection duration of the higher valent vaccination service reached the highest level, the market share still accounted for 93.0%, despite the distance to the vaccination site becoming longer and OOP payment becoming higher. The result of CV indicated that female students were willing to spend CNY 5612.79 to include Service 5 within their optional prevention program.

## 4. Discussion

Nowadays, female university students, as the beneficiaries of HPV vaccines, have a low vaccination rate [21]. Few studies have applied the DCE method to identify the factors influencing female university students’ preferences for HPV vaccination. In this study, the DCE method was used to quantify the preferences of female Chinese university students and explore its key factors so as to create a more informed policy design, provide more effective HPV vaccination services, and increase vaccination rates. Our study found that female students preferred vaccination services with greater effectiveness, a longer protection duration, and a shorter waiting time, as well as those at a longer distance and requiring lower OOP payment. This study not only provides reference for future adjustments in medical insurance policies but also supports the improvement and market development of HPV vaccination services.

Our findings suggest that female university students preferred vaccination services at a distance of more than 60 min, which is in contrast with the findings of another DCE study in Zhejiang [29]. This may be related to the allocation of vaccine resources, taking into account the fact that service points at longer vaccination distances are less densely populated, the population’s willingness to be vaccinated is not as high, and vaccine accessibility is better. However, most of the universities are in urban centers with high population density, where the appropriate vaccination population has a higher willingness to be vaccinated and female university students may have relatively less access to vaccination. Another potential explanation could be the social stigma or embarrassment associated with receiving an HPV vaccination. Students might prefer traveling farther to ensure privacy and avoid being recognized by peers. This aspect could be worth exploring to better understand the barriers to HPV vaccination among female university students and to inform more effective vaccination strategies. The results of this study can help policy makers to develop an appropriate policy on HPV vaccination, and achieve a rational allocation of HPV vaccine resources among regions. Meanwhile, the government can seek to achieve a wider government procurement of HPV vaccines on a district basis and increase access to vaccination for female university students, so as to increase the HPV vaccination rates among female students.

The heterogeneous nature of preferences is an important consideration for policymakers striving to improve the delivery of personalized vaccination services. Sex history, parents’ educational level, monthly living expenses, annual per capita household income, type of household registration, and university type were regarded as vital indicators of heterogeneity. The results of previous studies are consistent with our findings [27,28,29]. Relevant medical institutions should consider targeted vaccination schemes when formulating these programs.

We conducted a comprehensive analysis of the acceptability of four prevalent prevent services among female students, considering diverse price points and gauging the willingness of students to invest in upgrading HPV vaccination services. Our findings revealed that at the same distance, imported 9vHPV vaccination services emerged as the top choice for patients, closely followed by imported 4vHPV and domestic 2vHPV services, while imported 2vHPV services lagged behind as the least favored option. Notably, when the distance to vaccination was increased to 15–30 min away, the demand for high-valency vaccination services was the highest among students and exceeded the market share of imported 9vHPV vaccination services. Additionally, students exhibited a remarkable willingness to invest more than CNY 5612.79 to include additional vaccination service programs as part of their available options, underscoring the perceived value of incorporating more effective and longer-protection-duration HPV vaccination services as choices.

Moreover, we found that effectiveness is the most important attribute when deciding whether to receive the HPV vaccine, which is consistent with the results of a previous study in Zhejiang [29]. Female students tended to choose vaccination services with 90% protection effectiveness. At present, in addition to the 9vHPV vaccine, which has a better preventive effect, both 2vHPV and 4vHPV vaccines also exhibit excellent safety and immunogenicity. It has been reported that the 9vHPV vaccine covers all the HPV types that the 2vHPV and 4vHPV vaccines prevent, in addition to several other types, providing the most comprehensive protection against HPV-related diseases [46]. The appointment rate of the 9vHPV vaccine in the market was much higher than that of the 2vHPV and 4vHPV vaccines [47]. However, the vaccination rates of 2vHPV and 4vHPV remain lower in real world. Relevant pharmaceutical organizations may consider increasing the production and supply of the 9vHPV vaccine. The missionary department should strengthen education on 2vHPV and 4vHPV vaccines and improve the public’s awareness of the preventive effects of 2vHPV and 4vHPV vaccines. At the same time, it is also necessary to speed up the research on and development of other domestic high-valent HPV vaccines to achieve better preventive effects. This will help promote the prevention of cervical cancer in China and increase the HPV vaccination rate among female university students.

## 5. Limitations

This study has the following limitations. First, due to the geographic distribution of the university and the COVID-19 pandemic, the selected regions did not include China’s northeastern region, and so this work does not strictly cover the entire country. Therefore, the general applicability of these findings to the entire country must be interpreted with caution. Additionally, this study adopted a combination of online and offline face-to-face surveys. In the online survey, the students’ understanding of the questionnaire could not be distinguished, although a set of strict quality control standards ensured the quality of the questionnaire, and the results of the sensitivity analysis were reasonable (Appendix A).

## 6. Conclusions

This study explored the preferences for HPV vaccination services among female university students in China. Our findings reveal that the most critical factor influencing vaccine preference is effectiveness, followed by protection duration, OOP payment, and waiting time, while distance was considered less important. We conducted detailed analyses of preference patterns across various subgroups and examined how market shares for four prevalent vaccination services fluctuated at different price points. This analysis provides valuable insights that can inform future enhancements to personalized, need-based HPV vaccination services.

## Figures and Tables

**Figure 1 vaccines-12-00905-f001:**
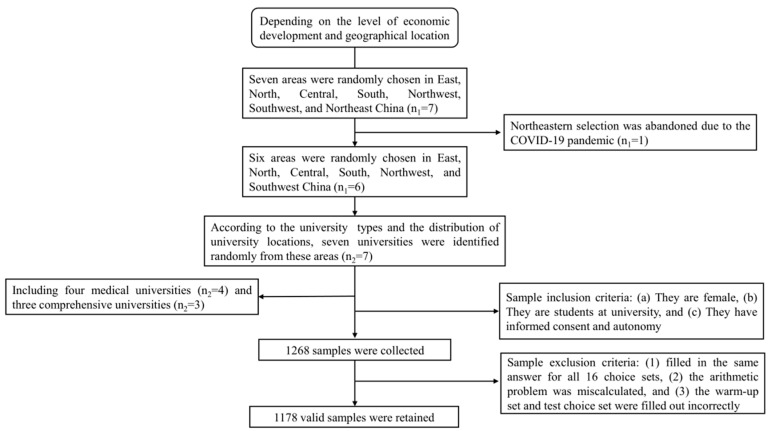
Sampling flowchart for research sites.

**Figure 2 vaccines-12-00905-f002:**
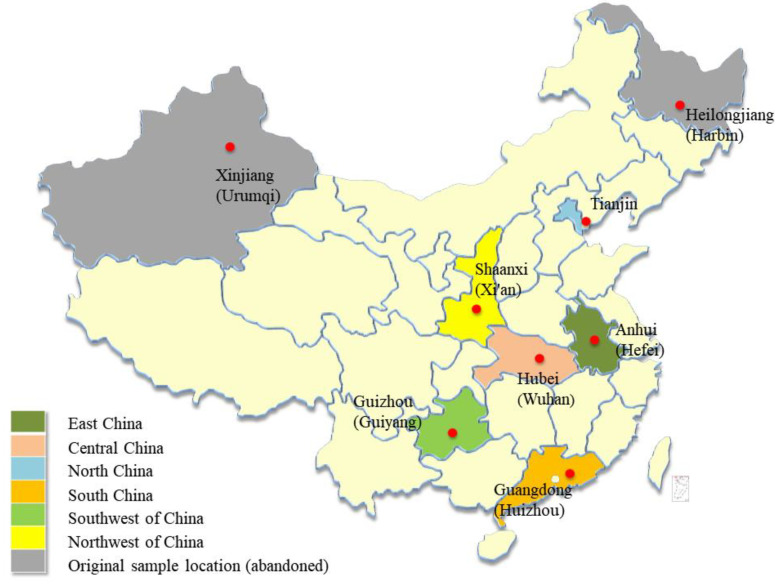
Locations of sampling.

**Figure 3 vaccines-12-00905-f003:**
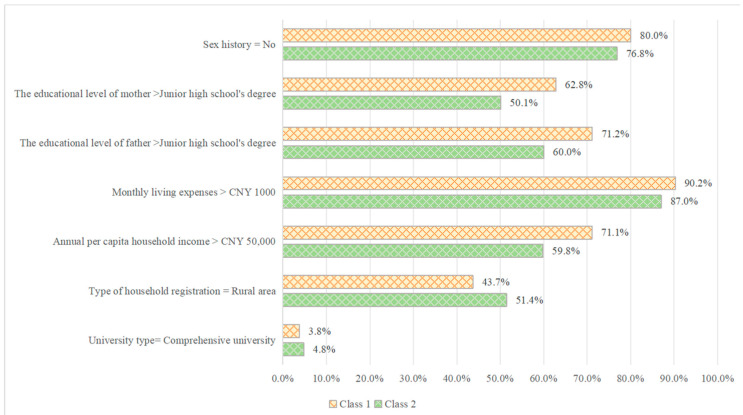
The profile of membership. Note: type of household registration is a public certificate book that registers and certifies a natural person according to their household, and records the natural person’s name, date of birth, relatives, marital status, etc. Currently there, are two main types of urban and rural household registration. CNY: Chinese Yuan.

**Figure 4 vaccines-12-00905-f004:**
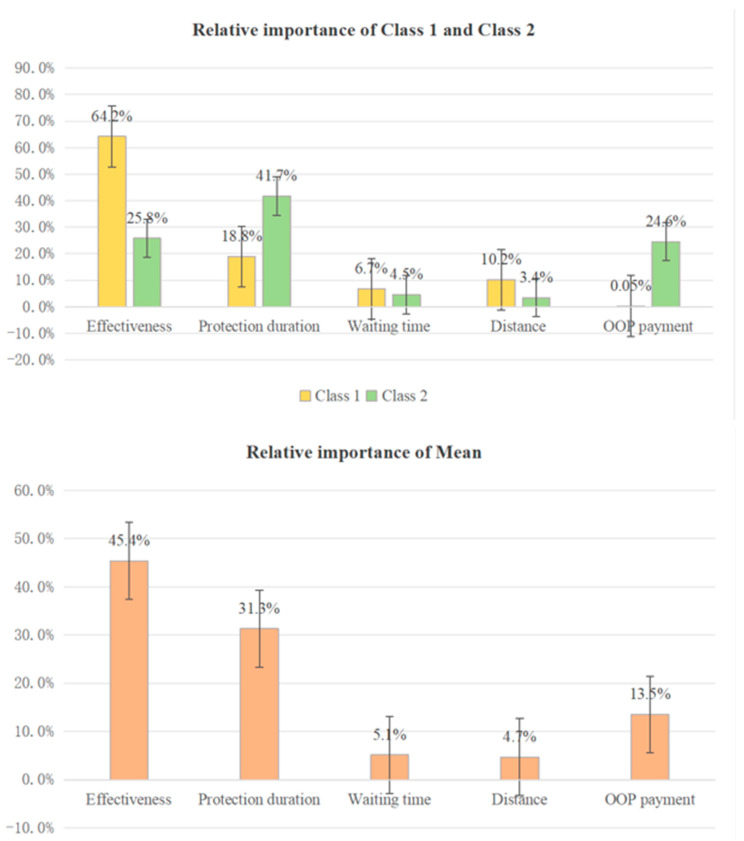
Relative importance of the attributes within each latent class and at the mean. Note: waiting time refers to the waiting time between appointment and successful vaccination of various types of cervical cancer vaccine; distance refers to the driving time it takes to reach the vaccination site by public transport; OOP: out-of-pocket.

**Table 1 vaccines-12-00905-t001:** Detailed definitions of attributes and levels.

Attribute	Definition	Attribute Level
Effectiveness	The prevention probability of vaccination against cervical cancer or genital warts.	A. Normal (it can prevent cervical cancer only; the prevention effectiveness is about 70%)
		B. Good (it can prevent cervical cancer and genital warts, the prevention effectiveness of cervical cancer is about 70%, genital warts are about 90%)
		C. Very good (it has preventive effect on cervical cancer and genital warts, and both the prevention effectiveness are about 90%)
Protection duration (year)	The duration of protection obtained after vaccination and describes the persistence of the vaccine’s effect.	A. 10
		B. 20
		C. 40
		D. Lifetime
Waiting time (month)	The waiting time between appointment and successful vaccination of various types of cervical cancer vaccine.	A. <3
		B. 3~6
		C. 6~9
		D. 9~12
Distance (minute)	The driving time it takes to get to the vaccination site by public transport.	A. <15
		B. 15~30
		C. 30~60
		D. ≥60
Out-of-pocket payment (CNY: Chinese Yuan)	Out-of-pocket (OOP) payment per person for full HPV vaccination *.	A. 1000
		B. 3000
		C. 5000
		D. 8000

Note: * this does not include the travel fee, the booking fee for manual renewal, etc., and only includes the total cost of three doses of vaccine.

**Table 2 vaccines-12-00905-t002:** Example of a choice set.

Attribute	Service A	Service B
Effectiveness	Very good ^a^	Normal ^b^
Protection duration (year)	40	40
Waiting time ^c^ (month)	6~9	6~9
Distance ^d^ (minute)	15~30	15~30
Out-of-pocket payment (CNY)	3000	3000
Which service would you choose?	□ Service A	□ Service B
	□ Neither (No preference for either and quit)	

Note: ^a^ indicates that it has a preventive effect on cervical cancer and genital warts, and both prevention effectiveness are about 90%; ^b^ indicates that it prevents cervical cancer only, and the prevention effectiveness is about 70%; ^c^ refers to the waiting time between appointment and successful vaccination of various types of cervical cancer vaccine; ^d^ refers to the driving time it takes to get to the vaccination site by public transport. CNY: Chinese Yuan.

**Table 5 vaccines-12-00905-t005:** The latent class logit model estimates (N = 1178).

Attributes	Levels	Class 1		Class 2	
		Est	SE	Est	SE
**Asc1**		2.456 ***	0.512	2.510 ***	0.122
**Asc2**		2.030 ***	0.519	2.518 ***	0.126
**Effectiveness**	Normal ^a^ **(ref)**				
	Good ^b^	2.710 ***	0.193	0.863 ***	0.060
	Very good ^c^	4.860 ***	0.241	1.498 ***	0.080
**Protection duration (years)**	10 **(ref)**				
	20	0.947 ***	0.116	0.913 ***	0.050
	40	1.091 ***	0.162	1.747 ***	0.081
	lifetime	1.426 ***	0.165	2.418 ***	0.086
**Waiting time ^d^ (months)**	<3 **(ref)**				
	3~6	−0.507 ***	0.129	−0.107 *	0.042
	6~9	−0.417 **	0.131	0.03	0.050
	9~12	−0.230 *	0.105	−0.233 ***	0.050
**Distance ^e^ (minutes)**	<15 **(ref)**				
	15~30	−0.302 **	0.100	0.099 *	0.049
	30~60	0.471 ***	0.084	0.002	0.048
	≥60	0.131	0.109	0.197 ***	0.046
**OOP payment (CNY)**		5.02 × 10^−7^ ***	0.000	−0.24 × 10^−4^ ***	0.000

Note: ^a^ prevents cervical cancer only and the prevention effectiveness is about 70%; ^b^ prevents cervical cancer and genital warts, the prevention effectiveness of cervical cancer is about 70% and that of genital warts is about 90%; ^c^ it has a preventive effect on cervical cancer and genital warts, and prevention effectiveness for both is about 90%; ^d^ refers to the waiting time between appointment and successful vaccination with various types of cervical cancer vaccine. ^e^ refers to the driving time it takes to reach the vaccination site by public transport; ref: reference; Est: estimate; SE: standard error; OOP: out-of-pocket. CNY: Chinese Yuan. * *p* < 0.05; ** *p* < 0.005; *** *p* < 0.001.

**Table 6 vaccines-12-00905-t006:** Uptake predictions of hypothetical scenarios (N = 1178).

Attribute	Service 1	Service 2	Service 3	Service 4	Service 5
Effectiveness	Normal	Normal	Good	Very good	Very good
Protection duration ^a^ (year)	10	10	20	40	Lifetime
Distance ^b^ (minute)	<15	<15	<15	<15	15~30
OOP payment (CNY)	1000	2000	3000	4000	6000
Choice set	Predicted uptakes			5th and 95th percentiles	
Service 1, 2, 3, and 4					
Pr (Choice = Service 1)	0.7%			(0.5~0.9%)	
Pr (Choice = Service 2)	0.6%			(0.5~0.8%)	
Pr (Choice = Service 3)	8.6%			(7.2~10.0%)	
Pr (Choice = Service 4)	90.1%			(88.4~91.7%)	
Service 1,2,3 and 5					
Pr (Choice = Service 1)	0.5%			(0.4~0.6%)	
Pr (Choice = Service 2)	0.4%			(0.3~0.6%)	
Pr (Choice = Service 3)	6.1%			(5.0~7.2%)	
Pr (Choice = Service 5)	93.0%			(91.6~94.3%)	
Choice set	Predicted uptakes			5th and 95th percentiles	
Service 1, 3 and 4 versus Service 1, 3, 4 and 5					
	5612.79			(4505.88, 6719.71)	

Note: ^a^ refers to the waiting time between appointment and successful vaccination with various types of cervical cancer vaccine. ^b^ refers to the driving time it takes to reach the vaccination site by public transport; OOP: out-of-pocket. CNY: Chinese Yuan.

## Data Availability

The raw data supporting the conclusions of this article will be made available by the authors without undue reservation.

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
