# Peer review of "Analyzing HPV Vaccination Service Preferences among Female University Students in China: A Discrete Choice Experiment"

_vaccines, 2024, doi:10.3390/vaccines12080905_

Round 1
Reviewer 1 Report
Comments and Suggestions for Authors
Hu et al. have investigated the preferences of female university students in China for HPV vaccination services through a discrete choice experiment (DCE). Their study aimed to explore the factors influencing students' decisions to use HPV vaccination services, potentially informing more effective vaccination strategies. They identified five key attributes: vaccine effectiveness, protection duration, waiting time, travel distance, and out-of-pocket costs. By analyzing responses from 1,178 valid questionnaires, they found that effectiveness was the most valued attribute, while travel distance was less concerning to the participants. Preferences emerged for vaccines offering 90% effectiveness, lifetime protection, a wait of less than three months, a travel time exceeding one hour, and minimal personal expense. The study also quantified students' willingness to pay for preferred service configurations, suggesting that significant modifications to current vaccination services could markedly increase uptake among this demographic. This research underscores the importance of tailored, need-based vaccination service designs to boost HPV vaccination rates among female university students in China.
The claims are properly placed in the context of the previous literature. The experimental data support the claims. The manuscript is written clearly enough that most of it is understandable to non-specialists. The authors have provided adequate proof for their claims, without overselling them. The authors have treated the previous literature fairly. The paper offers enough details of methodology so that the experiments could be reproduced.
Comments
1. The WHO Global Strategy aims to eliminate cervical cancer as a public health issue, with ambitious targets set for 2030 to guide global efforts. These include vaccinating 90% of girls with the HPV vaccine by age 15, screening 70% of women with a high-quality test by ages 35 and 45, and ensuring that 90% of women with diagnosed cervical disease receive treatment. This strategy seeks to reduce the incidence of new cervical cancer cases to fewer than 4 per 100,000 women annually. Given the context of your study, which focuses on HPV vaccination preferences among university students, referencing the WHO Global Strategy in your introduction could strengthen the manuscript by aligning it with international public health goals and highlighting the potential impact of increased vaccination uptake.
https://www.who.int/news-room/fact-sheets/detail/cervical-cancer
2. One of the survey results is intriguing: the preference of university students for vaccination services located more than 60 minutes away. This preference contrasts with trends in many Western countries, where the best coverage is achieved through school-based programs that administer vaccines directly in classrooms, eliminating the need for travel. The discussion addresses this unusual finding, suggesting it might relate to the allocation of vaccine resources and accessibility issues. However, another potential explanation could be the social stigma or embarrassment associated with receiving an HPV vaccination. Students might prefer traveling farther to ensure privacy and avoid being recognized by peers. This aspect could be worth exploring to better understand the barriers to HPV vaccination among female university students and to inform more effective vaccination strategies.
3. The authors have delineated various service alternatives based on the HPV vaccines currently available in the market, along with attributes such as the distance to the vaccination site. These include domestic and imported services for 2vHPV, 4vHPV, and 9vHPV vaccines, and a proposed 'Service 5' described as a higher valent vaccination service. However, it is important to note that, to the best of my knowledge, the highest valency HPV vaccine commercially available today is the 9vHPV vaccine, which protects against nine HPV types. Consequently, the 'Service 5' option, implying a vaccine covering more HPV types, does not currently exist as a commercial product. This raises concerns about the practical applicability of 'Service 5' as a realistic option for university students in China.
4. The term 'Service 5' in the abstract is currently vague, potentially leading to confusion. It would be beneficial to specify that 'Service 5' refers to a hypothetical vaccination service offering a vaccine with a higher valency than the currently available 9vHPV vaccine. This detail will help clarify the abstract and set realistic expectations for the readers. The revised sentence in the abstract could read:
'The result of the compensating variation (CV) analysis indicated that female students were willing to spend Chinese Yuan (CNY) 5612.79 to include a hypothetical higher valent vaccination service (beyond the 9vHPV vaccine) within their optional prevention program.'
This revision more accurately reflects the nature of 'Service 5' as a theoretical option not yet available in the market, enhancing the manuscript's clarity and factual correctness.
5. The authors write in the introduction and the discussion: 'These vaccines are highly effective in preventing infections and related diseases caused by the HPV virus, with the 9vHPV vaccine being the most effective and containing the types of HPV that can be prevented by the 2vHPV and 4vHPV vaccines'. This sentence may mislead readers by suggesting that the 9vHPV vaccine covers only the same types of HPV as the 2vHPV and 4vHPV vaccines. It is crucial to clarify the broader spectrum of HPV types covered by the 9vHPV vaccine. I recommend revising the sentence to reflect these distinctions clearly. For example: 'These vaccines are highly effective in preventing infections and related diseases caused by the HPV virus. The 2vHPV vaccine targets HPV types 16 and 18, which are responsible for approximately 70% of all cervical cancer cases. The 4vHPV vaccine additionally protects against HPV types 6 and 11, which cause around 90% of all genital warts cases. The 9vHPV vaccine further includes protection against HPV types 31, 33, 45, 52, and 58, collectively covering strains that cause approximately 90% of genital warts and cervical cancers.' This reformulation explicitly details the specific HPV types each vaccine covers, thereby enhancing the accuracy and usefulness of the information provided.
6. The submitted manuscript currently lacks line numbers. To enhance the accuracy of the peer review process and streamline subsequent edits, I strongly recommend implementing a line numbering system across all pages of the manuscript effective immediately. This change will significantly aid reviewers and editors in providing precise feedback.
7. In the manuscript, the authors have presented percentages with two decimal places. For ease of reading and to simplify the presentation in both tables and text, I recommend rounding these percentages to one decimal place (e.g., 92.90% to 92.9%).
8. MDPI Vaccines typically recommends that authors integrate figures and tables directly into the manuscript before submission. It complicates the review process when figures and tables are provided in a separate compressed zip file that must be downloaded independently of the main manuscript. For efficiency and ease of review, please embed these elements within the document itself.
9. In Table 1, there are several unconventional characters used to denote categories, which might confuse or distract readers. Common practice in scientific tables is to use standard alphanumeric characters or simple symbols that are universally recognized. I recommend replacing the special characters like ‘’, ‘‚’, ‘ƒ’, and ‘„’ with standard bullets, numbers, or letters. For example, you could use numerical or alphabetical ordering (1, 2, 3 or A, B, C) to clearly differentiate between the categories. This change would enhance the readability and accessibility of the table's content.
10. Typically, distance is quantified in units such as kilometers rather than time. In the manuscript, the term 'more than 60-minute distance' is frequently used without clarification on whether this refers to walking or driving time. Although Table 1 specifies that this measurement pertains to 'The driving time it takes to get to the vaccination site by public transport,' this critical detail should be consistently described throughout the manuscript to avoid confusion.
Minor revisions
Title, "Analyzing HPV Vaccination Preferences Among Female University Students in China: A Discrete Choice Experiment"
Abstract, "Objective: Despite being primary beneficiaries of human papillomavirus (HPV) vaccines, female university students in China exhibit low vaccination rates. This study aimed to assess their preferences for HPV vaccination services and evaluate the relative importance of various factors to inform vaccination strategy development.
Methods: Through literature review and expert consultations, we identified five key attributes for study: vaccine effectiveness, protection duration, waiting time, distance to vaccination site, and out-of-pocket (OOP) payment. A D-efficient design was used to create a discrete choice experiment (DCE) questionnaire. We collected data via face-to-face interviews and online surveys from female students across seven universities in China, employing mixed logit and latent class logit models to analyze the data.
Results: From 1,178 valid questionnaires, with an effective response rate of 92.9%, we found that effectiveness was the most significant factor influencing vaccination preference, followed by protection duration and OOP costs, with less concern for distance. The preferred attributes included a 90% effective vaccine, lifetime protection, a waiting time of less than three months, a travel time of more than 60 minutes, and low OOP costs. Significant variability in preferences across different vaccination service scenarios was observed, affecting potential market shares. The compensating variation (CV) analysis showed female students were willing to spend approximately CNY 5612.79 to include a hypothetical 'Service 5' (a vaccine with higher valency than the 9vHPV vaccine) in their prevention options.
Conclusion: The findings underscore the need for personalized, need-based HPV vaccination services that cater specifically to the preferences of female university students to increase vaccination uptake and protect their health."
Introduction, "Cervical cancer, primarily caused by the human papillomavirus (HPV) infection, remains a significant global health concern. Persistent infection with high-risk HPV types, especially type 16, is a known cause of cervical cancer. According to World Health Organization (WHO) data from 2020, there were approximately 604,000 new cases and 343,000 deaths due to cervical cancer worldwide. Notably, Chinese women accounted for about 110,000 of these new cases—representing approximately 18.3% of the global total—and around 60,000 deaths, or 17.3% of the worldwide fatalities."
Introduction, "These vaccines are highly effective in preventing infections and related diseases caused by the HPV types they cover. The 2vHPV vaccine targets HPV types 16 and 18, which are responsible for 70% of all cervical cancer cases. The 4vHPV vaccine additionally covers HPV types 6 and 11, accounting for 90% of all genital warts cases. The most comprehensive option, the 9vHPV vaccine, includes protection against HPV types 6, 11, 16, 18, 31, 33, 45, 52, and 58, collectively responsible for 90% of genital warts and cervical cancer cases."
Results, "Female students also had strong preferences for a service that requires more than a 60-minute drive by public transport to reach the vaccination site (β= 0.329 > 0.135, p < 0.001) and low out-of-pocket (OOP) payment."
Results, "Based on the HPV vaccines currently available in the market, and considering attributes including the distance to the vaccination site, we defined Service 1 as a domestic 2vHPV vaccination service, Service 2 as an imported 2vHPV vaccination service, Service 3 as an imported 4vHPV vaccination service, Service 4 as an imported 9vHPV vaccination service, and Service 5 as a hypothetical vaccination service offering a vaccine with higher valency than the 9vHPV vaccine."
Discussion, "It has been reported that the 9vHPV vaccine covers all the HPV types that the 2vHPV and 4vHPV vaccines prevent, in addition to several other types, providing the most comprehensive protection against HPV-related diseases."
Conclusion, "This study explored the preferences for HPV vaccination services among female university students in China. Our findings reveal that the most critical factor influencing vaccine preference is effectiveness, followed by protection duration, out-of-pocket payments, and waiting time, while distance was considered less important. We conducted detailed analyses of preference patterns across various subgroups and examined how market shares for four prevalent vaccination services fluctuated at different price points. This analysis provides valuable insights that can inform future enhancements to personalized, need-based HPV vaccination services."
As a non-native English speaker, I have suggested some reformulations to improve the language in the manuscript. However, I recommend considering a review by a native English speaker to further enhance sentence construction, readability, coherence, and accuracy, ensuring a polished final presentation.
Author Response
Dear editors and reviewers,
On behalf of my co-authors, we thank you very much for giving us an opportunity to revise our manuscript entitled “Preferences of Female University Students for HPV Vaccination Services in China: A Discrete Choice Experiment” (Manuscript ID: 3088542). We appreciate the constructive comments and suggestions that the editor and reviewers have provided on our manuscript.
We have studied the reviewers' comments carefully and have revised the manuscript accordingly, with changes marked in blue. We hope you find our revised-manuscript suitable for publication and look forward to hearing from you. Thank you and best regards.
Yours sincerely,
Lu Hu
Hulu89@stu.ahmu.edu.cn

Reviewer 2 Report
Comments and Suggestions for Authors
The manuscript essentially described a statistical method of eliciting choices for HPV vaccination amongst female university students in China.
It needs the following clarification.
1. The correct age group is 9 to 14 years for a national HPV vaccination program. Is it different in China? If so, can you state the reasons?
2. HPV Vaccination is a gender neutral vaccine. Why were only female students interviewed? And, why were only cervical cancer and genital warts used as diseases prevented by HPV Vaccination?
3. HPV causes cervical, vaginal and vulval cancer in women, anal and penile cancer in men, oropharyngeal cancer and genital warts in both men and women. Hence, it is no longer appropriate to target girls only for HPV Vaccination.
4. Targeting school children is the ideal age group suggested by WHO. Is this different in China? Why were 22 year old students included in the study?
5. How many students were excluded using the inclusion and exclusion criteria? Can you please give a decision tree or CONSORT diagram to explain?
6. What were the reasons for training students as interviewers?
7. What percentage of women students attend university in China? In other words, is the presumption that this sample of university women students appropriate?
8. There are several quaint English expressions that are not familiar to an average reader. These need to be thoroughly corrected.
9. There is excessive statistical formulae presented. Why do you think your method of sampling is superior to other methods available? After all, the choice of questions for the survey were determined by you as important or appropriate, not the women likely to receive the vaccination.
Comments on the Quality of English LanguageNeeds thorough revision with a native English speaking non-scientific, non-statistical person
Author Response

(The authors gave the same response as above.)

Round 2
Reviewer 2 Report
Comments and Suggestions for Authors
Thank you for answering to the queries raised by the reviewers . You have only defended3your statements, without any extra explanation.
Author Response
Dear Editors and Reviewers
Thank you for your letter and for the reviewers' comments concerning our manuscript entitled "Analyzing HPV Vaccination Services Preferences Among Female University Students in China: A Discrete Choice Experiment" (Manuscript ID:3088542). Those comments are all valuable and very helpful for revising and improving our paper, as well as the important guiding significance to our researches. We have studied comments carefully and have made correction which we hope meet with approval. These issues are not only clarified to the reviewer but also to the readers. In addition, we changed in order of authorship according to authors’ contributions. The paper was coauthored by Lu Hu, Jiacheng Jiang, Zhu Chen, Sixuan Chen and Xinyu Jin, Yingman Gao, Li Wang and Lidan Wang.
Thank you for your consideration. I look forward to hearing from you.
Sincerely,
Lu Hu
School of Health Management, Anhui Medical University, Hefei, China, 230032
E-mail: hulu89@stu.ahmu.edu.cn
